# Burnout and organisational stressors among healthcare staff working with adults with intellectual disabilities in Ireland

Patrick Clancy⬤*, Marica Cassarino⬤

School of Applied Psychology, University College of Cork, North Mall, Cork City, Ireland

* 117222539@umail.ucc.ie

## Abstract

### Background

The associations between organisational stressors and burnout among healthcare staff working with adults with intellectual disabilities are underexplored. This study investigated rates of burnout and associated stressors among Irish healthcare workers during the COVID-19 pandemic.

### Materials and methods

A convenience sample of 329 Irish frontline staff supporting adults with intellectual disabilities completed a survey assessing personal, work-related, and client-related burnout, and organisational stressors. Quantitative correlational analysis assessed bivariate and multivariate associations, while qualitative accounts were analysed thematically.

### Results

Compared to international data, we observed very high levels of personal and work-related burnout, with lower levels of client-related burnout. "Lack of resources" was the stressor holding the strongest association with burnout, followed by "bureaucracy" and "work-home conflict". Qualitative responses highlighted the negative impact of the pandemic on workload, service quality, and staff wellbeing.

### Discussion

Our findings highlight an important association between organisational stressors and burnout among frontline staff, suggesting the potential benefit of designing organisationally focused interventions to reduce stress and promote staff wellbeing.

**Data Availability Statement:** https://osf.io/ctpmb/.

**Funding:** The author(s) received no specific funding for this work.

**Competing interests:** The authors have declared that no competing interests exist.

## Introduction

Healthcare workers (HCWs) globally face many pressures at work and are at risk of burnout [1]. Given the interpersonal and emotional needs of individuals with intellectual disabilities, HCWs working in this area can be at a heightened risk of burnout due to the unique stressors that they may encounter both at an organisational and interpersonal level [2]. For instance, Robertson et al. [3] found that about a quarter of direct-care staff in England who worked with people with intellectual disabilities were significantly stressed, with a lack of resources and staff support outlined as the greatest sources of stress. Burnout is strongly associated with low ratings of client satisfaction, medical mistakes, and reduced sleep, it impacts not just the safety of clients, but also an organisation's long-term viability, by elevating staff turnover and decreasing revenue due to lower production [4].

Reports on the prevalence of burnout differ across population's worldwide [5]. This variability is possibly due to the lack of an accepted definition of burnout, and the subjectivity of its criteria of diagnosis [1]. The term burnout was first established by psychoanalyst Freudenberger to describe excessive demands being placed on an individual's resources, and energy, resulting in diminished vitality [6]. However, defining burnout has proven problematic, these difficulties have likely hampered the gathering of knowledge and the treatment of burnout as a result [7]. The World Health Organisation describes occupational burnout as a persistent psychological syndrome, brought about by subjection to consistent emotional and interpersonal stress generated by the work, or in the workplace [8]. A similar definition of burnout was provided by an international panel of experts who suggested that "occupational burnout or occupational physical AND emotional exhaustion state is an exhaustion due to prolonged exposure to work-related problems" [9, p95].

The variability in defining burnout also transpires in measurement. Maslach's Burnout Inventory (MBI) [10] is the most frequently used scale when measuring levels of burnout, with evidence coming from studies involving nurses [11, 12], social workers [13], and staff working with individuals with intellectual disabilities [14]. Maslach & Leiter [15] use the three aspects of (i) emotional exhaustion, (ii) depersonalisation and (iii) personal accomplishment to define burnout. This differs from Guseva et al.'s [9] definition, in which physical and emotional exhaustion is emphasised while depersonalisation and personal accomplishment were not included. While Maslach et al.'s [10] operationalisation is more comprehensive than others, it has been argued by Kristensen et al. [16] that fatigue and exhaustion encapsulate the primary aspects of burnout, and that depersonalisation and personal accomplishment should be excluded from the measurement of burnout, as these are separate phenomena that should be understood independently. Kristensen et al. [16] proposed the Copenhagen Burnout Inventory (CBI) as a measurement centred on fatigue and exhaustion; the CBI considers three subdimensions of personal burnout (i.e., the level of physical and psychological burnout felt by the individual), work-related burnout (i.e., the level of psycho-physical fatigue and exhaustion linked to an individual's work), and client-related burnout (i.e., the degree of physical and psychological exhaustion linked to working with clients). Studies among HCWs have found levels of CBI personal burnout to be the highest, and levels of CBI client-related burnout to be the lowest [16, 17].

A recent systematic review compared studies using the CBI and MBI to estimate the prevalence of burnout in HCWs and students [18], and found that there has been an increase in the use of CBI since 2016 across a variety of countries, thanks to the easier language and formulation of questions than in the MBI; this has been linked to higher response rates and use in countries beyond Europe (where the MBI has most been used). Importantly, the CBI appears to identify higher rates of burnout than the MBI (53% vs. 35%), with prevalence in line with

those found in a separate systematic review [19] for CBI among midwives. As the CBI is a more recent measure than the MBI, further research is needed to better understand the prevalence of burnout, and the associated stressors, in different contexts.

Studies support a strong association between burnout and workplace stress [15, 20, 21] however, there is a lack of knowledge as to the specific organisational stressors driving personal, work-related, and client-related burnout among HCWs working with adults with intellectual disabilities. A consideration of organisational stressors is justified by evidence that, while individual interventions to reduce burnout among HCWs can have a positive impact [22], organisationally directed interventions yielded superior treatment success [20, 21, 23–25]. Interventions which engage in structural alterations, promote greater correspondence between team members, and emphasise team collaboration, communication, and autonomy, can have great success in reducing burnout [26]. It was shown convincingly by Le Blanc & Schaufeli [27] that an organisational based team intervention could be effective in reducing levels of burnout. Given the impact organisational interventions have had in reducing burnout, understanding the associations between burnout and different organisational stressors is critical in tackling burnout. To this end, Vassos & Nankervis [28] examined factors associated with burnout among staff supporting individuals with intellectual disabilities, by organising the Staff Stressor Questionnaire (SSQ) [29] subscales into four "approaches": Individual, organisational, interpersonal, and demographic. The authors examined associations between these four dimensions and burnout using the MBI burnout scales [10], and found that many of the subscales of the SSQ had medium to large statistically significant associations with the MBI subscales [28]. However, an association between SSQ dimensions and the CBI has not been explored yet.

It has been widely reported that the levels of burnout among frontline healthcare workers have significantly increased in recent years [30, 31], and the COVID-19 pandemic has put further strains on HCWs' wellbeing. A scoping review [32] found increased reports of stress, anxiety, and depressive symptoms in this population. Studies indicate that levels of burnout among frontline HCWs such as nurses has significantly increased since the Covid-19 Pandemic [30, 33]. HCWs were exposed to personal, organisational, and cultural stressors, such as the possibility of infection, and staff redeployment, which have increased the risk of burnout [34]. Based on these findings, it is important to consider the multiple organisational stressors that may be linked to burnout in light of the pandemic, as these can provide useful insights for healthcare practice on how to reduce overall levels of burnout among staff as well as how to address and cope with future emergency situations, particularly given the predicted likelihood of another deadly pandemic occurring [35].

Limited research has been conducted on how burnout levels among HCWs supporting people with intellectual disabilities have been impacted by the COVID-19 pandemic. This topic is particularly important in countries such as Ireland, where caring for individuals with intellectual disabilities constitutes an important part of healthcare work. The 2016 Irish Census indicated that 1.4% of the Irish population (66,611 people) reported intellectual disabilities, with 19% of these living full-time in community group homes or residential care centres [36]. A previous study investigated rates of burnout for Irish-based HCWs working with intellectual disabilities using the CBI scale [37] and found slightly higher levels of burnout (personal burnout- 57.92, work-related burnout- 55.10) than those found in the meta-analyses by Alahmari et al. [18] and Suleiman-Martos et al. [19]. This may indicate that Irish frontline staff working in this sector may be in greater jeopardy of developing burnout than in other healthcare professions, and HCWs working with adults with intellectual disabilities in other countries. However, that study did not consider the potential organisational stressors associated with the three

dimensions of burnout, which warrants a new investigation to identify the most important drivers or stress and burnout.

## Study aims

To address the gaps discussed above, this study aimed to investigate how organisational factors relate to personal, work -related, and client-related burnout among Irish-based HCWs supporting adults with intellectual disabilities, by examining associations between burnout dimensions assessed in the CBI and relevant organisational stressors measured via the SSQ. This will provide an insight regarding the levels of burnout among Irish HCWs supporting adults with intellectual disabilities, and the organisational stressors driving their burnout, the large variance in levels of burnout globally points to the need of investigating variations in burnout across different geographical and cultural contexts [5].

Vassos & Nankervis's [28] method of fitting SSQ subscales into individual, organisational, and interpersonal dimensions, and examining associations with the MBI subscales of emotional exhaustion, depersonalisation and personal accomplishment, was a hugely proficient method of shedding light on the drivers of organisational burnout; however, it is felt that directly examining how organisational stressors impact personal burnout, work-related burnout and client-related burnout may be of even greater utility in providing an insight into the organisational stressors driving different types of burnout.

The study tested six hypotheses.

(H1) Irish-based HCWs would report highest levels of personal burnout, than work-related burnout, and lowest levels of client-related burnout. This hypothesis was formulated from the PUMA study (Project on Burnout, Motivation and Job Satisfaction that was used to analyse the validity and reliability of the CBI) [16], as well other literature reviewed [17].

(H2) HCWs would report that the COVID-19 pandemic has increased their levels of stress, in line with accumulating evidence of elevated levels of burnout among HCWs during the Covid-19 pandemic [30, 31, 38], as well as the increased stressors during the Covid-19 pandemic [34].

(H3) Personal burnout would have the strongest association with the SSQ subscale of "lack of resources". This hypothesis is made on account of Vassos & Nankervis [28] placing the SSQ subscale "lack of resources" into the category of "individual approach" (which has similarities to personal burnout), and the finding that it had the strongest association with the MBI subscale of "emotional exhaustion", which is how the CBI measures burnout.

(H4) Work-related burnout would have the strongest association with the SSQ subscale of "low status job". This hypothesis is made due to Vassos & Nankervis [28] placing the SSQ subscales of "bureaucracy" and "low status job" into the category of "organisational approach" (which is comparable to work-related burnout), and the identification that "low status job" was the SSQ subscale with the strongest association with the MBI subscale of "emotional exhaustion".

(H5) Client-related burnout would have the strongest association with the SSQ subscale of "client challenging behaviour". This hypothesis is made as a result of Vassos & Nankervis [28] fitting the SSQ subscale "client challenging behaviour" into the "interpersonal approach" category (which has similarities to client-related burnout), and the finding that it has the strongest correlation with the MBI subscale of "emotional exhaustion".

(H6) The participants of this study will report a higher level of burnout in all three categories than the average social care worker in the PUMA study. This hypothesis was made due to the increased levels of CBI burnout found among HCWs during the Covid-19 pandemic [30, 31], as well as McMahon et al. [38] findings of higher than levels of burnout among an Irish

sample of staff supporting individuals with intellectual disabilities, than other literature examined [18, 19].

## Materials and methods

### Design

This study used a cross-sectional survey that examined how staff stressors associate with personal, work-related, and client-related burnout. An online survey was used to collect both quantitative and qualitative data from participants; this method was chosen over in-person data collection as it allowed access to large and diverse range of prospective participants in a time efficient manner [39], and maximising reach, as our data collection was at a time when some level of Covid-19 related restrictions was still in place in healthcare settings. Online surveys also offer participants as much time as needed to answer questions, and they can complete the study when it is convenient to them to do so [40].

Both quantitative and qualitative data was collected and analysed as part of this study to have a comprehensive understanding of the experiences of stress and burnout among participants [41]. The core component of this study was quantitative, while the supplemental component was qualitative, this methodology is described by Schoonenboom & Johnson [41] as a deductive-simultaneous design.

### Participants

Participants of this study included people aged 18 years or older, and who were employed as healthcare staff supporting adults with intellectual disabilities as part of an organisation, at the time of data collection. Respondents to this study participated voluntarily. Purposive convenience sampling was employed as the sample needed to address the research hypotheses was specific but participants selected themselves based on their availability and accessibility to the survey [42]. This method was also chosen due to constraints in time and resources needed for probability sampling approach. Participants were recruited through an invitation poster that was distributed via Irish Social Care Worker social media platforms, as well as by email via the researchers' personal network, and through posters displayed in the work settings of staff who work with adults with intellectual disabilities. 371 valid responses were collected, all 371 of these participants began and completed all demographic questions and Copenhagen Burnout Inventory (CBI) subscale items. 329 of these participants (88.68%) began the Staff Stressor Questionnaire (SSQ) and completed all SSQ items, leaving a final sample of 329 participants.

The study received ethical approval by the School of Applied Psychology Research Ethics Committee, University College Cork (Reference number: EA-MMH01132022336) in January 2022. All participants read an information sheet which informed about the nature of the study, confidentiality, and right to withdraw. Participants were asked to provide written consent prior to taking part in the study, in line with institutional requirements and good practice on conducting online surveys [40].

### Materials and measures

The survey consisted of four sections, the first three of which were quantitative. The first section measured burnout through the CBI. The second segment was work stressors, measured using the SSQ. The third part of the survey measured covariates. The fourth and final part of the survey included two qualitative questions.

**Burnout.** The Copenhagen Burnout Inventory (CBI) was used to measure levels of burnout among the participants [16]. The CBI defines burnout as physical and psychological

fatigue. Personal burnout describes burnout experienced by the individual, work-related burnout is connected to work, and client-related burnout is burnout judged by the individual to be linked to their work with clients. The CBI is a 19-item questionnaire in which participants respond on a 5 point- Likert scale, that included the responses of either "Always"- 100, "Often"- 75, "Sometimes"- 50, "Seldom"- 25, "Never/almost never"- 0, or "To a very high degree"- 100, "To a high degree"- 75, "Somewhat"- 50, "To a low degree"- 25, "To a very low degree"- 0. The total score on this scale is the average of the scores on these items [16]. Each of these subscales' measures burnout from 0–100, a score of 50–74 is considered a moderate level of burnout, a score of 75 to 99 is considered a high level of burnout, while a score of 100 meets the criteria for severe burnout [38]. Personal burnout consists of the first 6 items. Work-related burnout contains the next 7 items (the last item reversed scored). Client-related burnout consists of the final 6 items.

The three scales of CBI personal, work-related, and client-related burnout have been found to be valid and reliable measures of burnout [16, 37]. The CBI scales of personal, work-related, and client-related burnout have previously been used to measure burnout among Irish HCWs working in mental health services during the Coronavirus pandemic [17], and were found to have a high degree of internal consistency with a Cronbach's Alpha of α = 0.89 for total CBI [43]. A Cronbach Alpha's of α = 0.85–0.87 was found among the subscales in the PUMA study, indicating very high internal reliability [16]. High levels of internal consistency were also found in the CBI subscales for this study, with a Cronbach's alpha of α = 0.85 found for personal burnout, α = 0.88 found for work-related burnout, and α = 0.9 found for client-related burnout [16].

**Work stressors.** The Staff Stressor Questionnaire (SSQ) is a 33- item questionnaire that measures how stressful potential stressors may be for the staff who support adults with intellectual disabilities [29]. The measure is comprised of 7 factors, (i) client challenging behaviour- 9 items- these items relate to how challenging behaviour impacts staff and the service, (ii) poor client skills- 7 items- these items describe a deficiency of client skill, and level of care staff need to provide, (iii) lack of staff- 3 items- these items relate to deficient support from co- workers, supervisors, and management, (iv) lack of resources- 3 items- this references insufficient availability of physical and staff resources, (v) low status job- 5 items- items relate to being employed in a low status job, with poor pay and few opportunities of promotion, (vi) bureaucracy- 3 items- items pertain to excessive complication regarding rules, regulations, and paperwork, (vii) work-home conflict- 3 items- items relate to friction between the staff members home and work life [29]. Participants who score higher signal that they endure a greater level of that stressor, in the workplace.

An acceptable level of internal consistency was reported by Hatton et al. [29] that ranged from α = 0.58–0.89, as well as good face, construct, and criterion-related validity. While no previous studies have analysed the psychometric properties of the SSQ scales in the Irish context, in line with Hatton et al. [29], in the present study, varying levels of internal consistency were also found in the SSQ subscales, with a Cronbach Alpha's of α = 0.89 for "client challenging behaviour", α = 0.84 for "poor client skill", α = 0.75 for "lack of staff support", α = 0.6 for "lack of resources", α = 0.7 for "low status job", α = 0.59 for "bureaucracy", α = 0.56 for "work-home conflict". A Cronbach's alpha of 0.7 or greater is customarily acknowledged as a benchmark of acceptable internal consistency, however, lower values may be judged as acceptable when a small number of items are contained in the scale [44]. Although Devereux et al. [45] removed 4 items to avoid overlap with their staff support measure, the remaining items of the Staff Stressor Scale were found to have good reliability, exhibiting a Cronbach's alpha of .90 for their sample of staff.

**Covariates.**   Covariates for this study included socio-demographic questions and further questions related to work circumstances.

Socio-demographic questions included participant's gender, age, level of education, duration of employment in their current role.

Work circumstances were investigated with questions related to the number of hours typically worked in a week in this role, their current work setting (day service, residential service, both, or other), and the frequency with which the participant had worked in the past six months with service uses with intellectual disabilities who also have mental health difficulties (responses on a 5-point Likert scale from "never" to "all the time").

We also asked participants to rate how they felt that the COVID-19 pandemic had influenced their level of stress at work, with responses on a 5-point Likert scale from "much worse than before" to "much better than before".

**Qualitative questions.**   The survey closed with two qualitative questions to (a) further explore the subjective experience of work-related stress associated with the Covid-19 pandemic, and (b) give the participant an opportunity to add comments related to their responses to the quantitative questions or further input on the topic.

## Data collection procedure

Data was collected online between the dates of January the 29th 2022 and April the 21st 2022. Literature has detailed the importance of carrying out a pilot study for online surveys, as a means of verifying the appropriateness and order of the questions, the comprehensiveness of the study, that instructions provided in the study are clear and user-friendly, and to confirm that technology involved is working correctly [39]. Before data collection in this study, the survey was piloted with five individuals who gave feedback on the quality of the questions and any technical issues, however given that no changes to the study were required, these responses were included in the data analyses.

## Data analysis

The collected data was downloaded from Qualtrics to excel for an initial inspection and cleaning. IBM SPSS 28.0 statistical software [46] was used to carry out the quantitative analysis, while NVivo v.12 [47] was used for qualitative data.

**Quantitative data analysis.**   The descriptive statistics for the quantitative data was presented as mean (M) and standard deviation (SD) for continuous variables, for categorical measures it was as percentages and frequencies. Cronbach's Alpha enables the assessment of the internal consistency of a scale [48], when test items are correlated to one another the alpha value is elevated, however if the length of the test is short the value of alpha is lessened [49]. The internal consistency of the CBI personal, work-related, and client-related burnout scales was analysed using Cronbach's alpha. The three outcome measures of personal, work-related, and client-related burnout, all showed a significant departure from normality ($p < .001$). Central Limit Theorem (CLT) indicates that larger samples form their own normal distribution, although in this study random sampling was not employed, it was assumed that a large sample of 371 would approximate appropriately to a normal distribution [50]. However, as Sawada [51] points out, the assumption of normality is rarely assured by CLT. Pearson's correlation coefficient was used to analyse associations between the outcome variables (CBI burnout subscales) and the predictor variables (SSQ subscales). Cohen [52]'s conventions were used to interpret effect size, with a correlation coefficient of 0.1 judged to represent a small association, a correlation coefficient of 0.3 was deemed a moderate correlation, and a correlation coefficient of 0.5 or larger was considered a large coefficient.

A one sample t-test was carried out to compare the mean scores for CBI personal, work-related, and client-related burnout in this sample, with the means observed in the PUMA study. A one sample t-test is a statistical hypothesis tool which can be utilised to assess the mean difference of a sample and a population mean [53].

A hierarchical-multiple regression was carried out for personal, work-related, and client-related burnout, in order to estimate the association between stressors and burnout before and after controlling for covariates, in line with the approach taken in previous research [54]. Each hierarchical multiple regression contained three steps, the first included the predictor variables of the SSQ subscales, the second step included the covariates of demographic factors, the third step included the covariates of the influence of the Covid-19 pandemic on stress levels and having worked with a client with mental health difficulties within the last 6 months. A standard confidence interval of 95%, and a p value = 0.05 was employed to decide on statistical significance [55].

**Qualitative data analysis.** Thematic analysis was used to analyse the qualitative data. Thematic analysis identifies, analyses, and reports on themes and patterns from the data [56]. A theme conveys some level of patterned response or meaning from the data. It is noted by Braun & Clarke [57], that themes do not passively emerge from the data, and that themes are better described as interpretive and creative stories about the data, generated by the researcher's theoretical assumptions, analytic skill and the data itself. The six steps outlined by Braun & Clarke [56] were used to conduct this thematic analysis. An inductive approach was primarily leaned on to code the data, in which the themes identified had strong links to the data. Themes were identified on a semantic level in which the explicit meaning of the data was organised to reveal patterns, and were then interpreted [56]. In approaching the qualitative data, considerations about reflexivity were made in relation to the main author's (PC) positionality in this research study. PC was at the time of writing a postgraduate student with considerable experience working as a social care worker with adults with intellectual disabilities, both in residential and day-service settings, and bringing the personal experience of having family members with intellectual disabilities. Reflecting on these 'insider' experiences, and discussing this with the lead author (MC), who brought an 'outsider' perspective, enabled a grounded approach to the data collection, analysis and interpretation. For example, the addition of qualitative questions to the survey was a result of this shared reflection to have a more nuanced understanding of the quantitative data. The methodology of the open- ended question regarding "the influence of the Covid-19 pandemic on stress levels", was deemed appropriate due to novel nature of the pandemic, and so that a direct insight into the Covid-19 stressors among staff working with adults with intellectual disabilities may be obtained. An open- ended question of "any additional comments?" was added so that the participants could include any further information that they felt was relevant. Due to the limited size of the qualitative data (i.e., the open-ended responses in the survey tended to be short), it was felt that identifying themes at an explicit semantic level was suitable, with an inductive approach so that themes which were identified had strong links to the data.

## Results

### Descriptive statistics

**Sample characteristics.** The main sample characteristics are reported in S1 Table. Women comprise the majority of the sample (91%) and 61% of respondents reported being under the age of 40 years old. 95.2% of the sample had complete undergraduate or postgraduate education and 79% of participants reported having at least 3 years of work experience. 77.4% of the respondents detailed working over 34 hours a week. About two thirds of

respondents (58%) worked in residential settings, while 23% worked in day care settings, 14% in both, and 5% in other types of settings, such as respite and community based services. Except for two respondents who worked in service management, all the participants reported working directly with clients and 83% reported having worked often or all the time with individuals with intellectual disabilities who also experienced mental health difficulties. 83.5% of respondents felt that the Covid-19 pandemic had made their level of work-related stress worse or much worse than before.

**CBI burnout and staff stressors.** All 371 participants completed all personal, work-related, and client-related CBI items, the scores are reported in Table 1. The mean levels of personal and work-related burnout were found to be in the "moderate" range. The mean of client-related burnout was below the "moderate" range.

The scores of the SSQ Staff Stressor Questionnaire are reported in S2 Table. "Lack of resources" was found to have the highest mean, while "poor client skill" had the lowest mean.

## Comparing levels of burnout from this study with the PUMA study

A one-sample t-test was run to determine whether burnout scores from this sample were different to levels from social care workers from the PUMA Study. The mean burnout score for personal burnout from this study (M = 65.67, SD = 15.05) was notably higher than the mean PUMA Study personal burnout mean score among social care workers (M = 38.7), with a statistically significant mean difference of 26.97, 95% CI [25.43 to 28.5], t(370) = 34.52, p < .001, Cohen's d = 1.79. The mean burnout score for work-related burnout from this study (M = 64.28, SD = 17.64) was considerably higher than the PUMA Study work-related burnout mean score among social care workers (M = 34.6), there was a statistically significant mean difference of 29.68, 95% CI [27.87 to 31.48], t(370) = 32.4, p < .001, Cohen's d = 1.68. The mean burnout score for client-related burnout from this study (M = 43.59, SD = 21.79) was higher than the PUMA Study client-related burnout mean score among social care workers (M = 34.1), with a statistically significant mean difference of 9.49, 95% CI [7.26 to 11.71], t(370) = 8.39, p < .001, Cohen's d = 0.44.

## Pairwise correlations

Pearson's correlation coefficient was used to analyse associations between personal, work-related, and client-related burnout, and the SSQ subscales (see S3 Table). All measures were

**Table 1. CBI burnout.**

|  | N | % | M | SD |
|---|---|---|---|---|
| Personal burnout | 371 |  | 65.67 | 15.05 |
| Moderate Personal burnout | 241 | 65 |  |  |
| High Personal burnout | 69 | 18.6 |  |  |
| Severe Personal burnout | 3 | 0.8 |  |  |
| Work- related burnout | 371 |  | 64.28 | 17.64 |
| Moderate Work- related burnout | 192 | 51.8 |  |  |
| High Work- related burnout | 96 | 25.8 |  |  |
| Severe Work- related burnout | 1 | 0.3 |  |  |
| Client- related burnout | 371 |  | 43.49 | 21.79 |
| Moderate Client- related burnout | 111 | 29.9 |  |  |
| High Client- related burnout | 21 | 5.7 |  |  |
| Severe Client- related burnout | 3 | 0.8 |  |  |

positively correlated with each other with moderate to strong correlations, showing that HCWs experiencing higher levels of stressors also reported higher levels of burnout. Considering the strongest associations, both personal and work-related burnout had a strong positive correlation with lack of resources, whereas client-related burnout had the strongest association with client challenging behaviours. Of note, personal and work-related burnout showed a very high correlation (r = 0.80, p < .01) which may indicate some potential overlap between the two constructs.

## Hierarchical multiple regression

Multiple linear regression was employed to identify the strongest predictors of burnout among the staff stressors, while controlling for demographic as well as "impact of Covid-19 pandemic", and "mental health difficulties", separately for personal burnout (Table 2), work-related burnout (Table 3), client-related burnout (Table 4).

Considering personal burnout, the analyses showed that after controlling for covariates, "lack of resources" was the strongest predictor ($b = 11.36$, $p < .001$), followed by "bureaucracy" ($b = 4.71$, $p < .05$). No other stressors or covariates had a significant effect on burnout.

Similar effects were found for work-related burnout, with "lack of resources" being the strongest predictor of increased levels of burnout, followed by "bureaucracy", across all three models (respectively, b = 17.69, p < .001; b = 6.43, p < .01). In this case, however, "lack of staff support" emerged also as a significant predictor when controlling for covariates (b = 4.21, p < .05).

Lastly for client-related burnout, "lack of resources" was once again the strongest predictor of increased levels of burnout across all three models (b = 7.58, p < .05), followed by "poor client skill" (b = 4.94, p < .001 in Model 3) and "client challenging behaviour" (b = 4.59, p < .001). None of the other stressors or covariates had a statistically significant effect.

**Table 2. Regression: Personal burnout.**

| Measure | Model 1 | | Model 2 | | Model 3 | |
|---|---|---|---|---|---|---|
| | **B** | **$R^2$** | **B** | **$R^2$** | **B** | **$R^2$** |
| SSQ Lack of resources | 12.03*** | 0.35 | 11.51*** | 0.37 | 11.36*** | 0.37 |
| SSQ Client challenging behaviour | 0.82 | | 0.7 | | 0.69 | |
| SSQ Lack of staff support | 1.61 | | 2.24 | | 2.23 | |
| SSQ Low job status | 1.3 | | 1.4 | | 1.4 | |
| SSQ Bureaucracy | 4.51* | | 4.74* | | 4.71* | |
| SSQ Poor client skills | 1.3 | | 0.35 | | 0.34 | |
| SSQ Work- home conflict | 3.95* | | 3.15 | | 3.08 | |
| Age | | | -1.27 | | -0.72 | |
| Gender | | | 19.42 | | 20.09 | |
| Level of education | | | -14 | | -14.02 | |
| Duration of employment | | | 2.66 | | 2.66 | |
| Hours worked in a week | | | 9.89 | | 9.42 | |
| Work setting | | | -5.27 | | -5.16 | |
| Mental health difficulties (client) | | | | | 3 | |
| Covid- 19 pandemic | | | | | 0.13 | |

Notes: Statistical significance is presented as

* p < .05

** p < .01

*** p < .001

**Table 3. Regression: Work-related burnout.**

| | Model 1 | | Model 2 | | Model 3 | |
|---|---|---|---|---|---|---|
| Measure | B | $R^2$ | B | $R^2$ | B | $R^2$ |
| SSQ Lack of resources | 18.85*** | 0.46 | 17.9*** | 0.47 | 17.69*** | 0.47 |
| SSQ Client challenging behaviour | 1.52 | | 1.36 | | 1.38 | |
| SSQ Lack of staff support | 3.04 | | 4.37* | | 4.21* | |
| SSQ Low job status | 2.78 | | 2.84 | | 2.9 | |
| SSQ Bureaucracy | 6.07* | | 6.31* | | 6.43** | |
| SSQ Poor client skills | -0.05 | | 0.37 | | 0.28 | |
| SSQ Work- home conflict | 5.59* | | 4.07 | | 4.02 | |
| Age | | | -1.74 | | -1.85 | |
| Gender | | | 24.5 | | 24.74 | |
| Level of education | | | -9.25 | | -9.95 | |
| Duration of employment | | | 0.32 | | -0.73 | |
| Hours worked in a week | | | 13.98 | | 13.49 | |
| Work setting | | | 6.92 | | 6.46 | |
| Mental health difficulties (client) | | | | | -1.23 | |
| Covid- 19 pandemic | | | | | -3.77 | |

Notes: Statistical significance is presented as

* p < .05

** p < .01

*** p < .001

**Table 4. Regression: Client-related burnout.**

| | Model 1 | | Model 2 | | Model 3 | |
|---|---|---|---|---|---|---|
| Measure | B | $R^2$ | B | $R^2$ | B | $R^2$ |
| SSQ Lack of resources | 6.46* | 0.31 | 7.58* | 0.32 | 7.58* | 0.32 |
| SSQ Client challenging behaviour | 4.54*** | | 4.6*** | | 4.59*** | |
| SSQ Lack of staff support | 0.29 | | -0.34 | | -0.3 | |
| SSQ Low job status | 0.24 | | 0.61 | | 0.47 | |
| SSQ Bureaucracy | 4 | | 2.02 | | 4.12 | |
| SSQ Poor client skills | 5.2*** | | 4.92*** | | 4.94*** | |
| SSQ Work- home conflict | -1.49 | | -1.17 | | -1.19 | |
| Age | | | -5 | | -1.63 | |
| Gender | | | -1.85 | | -4.73 | |
| Level of education | | | -4.84 | | -4.63 | |
| Duration of employment | | | 7.89 | | 8.21 | |
| Hours worked in a week | | | -13.53 | | -13.57 | |
| Work setting | | | 9.17 | | 9.35 | |
| Mental health difficulties (client) | | | | | 1.67 | |
| Covid- 19 pandemic | | | | | 1.22 | |

Notes: Statistical significance is presented as

* p < .05

** p < .01

*** p < .001

## Qualitative responses

A far greater number of participants (n = 248, 75.38% of the final sample) responded to the qualitative question "In what ways has the Coronavirus Pandemic impacted your level of stress?" than responded to the second qualitative question: "Do you have any additional comments?" (n = 95, 28.88% of the final sample). Thematic analysis was used to analyse this data, specifically the six steps outlined by Braun & Clarke [56]. A variety of strong themes were discerned from the data.

**Excessive workload.**    The primary theme that was detailed in the data was the "excessive workload" that was brought about by the Coronavirus Pandemic. Extra administrative paperwork, cleaning requirements, and staff shortages were outlined as the primary sources of this additional workload. Staff described frustration with these tasks having to be prioritised over the needs of service users. "The pandemic has excelled staffs stress levels, we have a mountain of extra paperwork to get through during shift because of it. We have so many extra duties to do, that takes time away from us that we used to spend supporting the residents" (ID: 350). It was outlined by some participants that it was the increased workload and a shortage of staff, driving staff burnout, as opposed to the service users being supported. "Burn out is not down to the residents, its the lack of support from the general staff shortages and now the increased workload" (ID: 282).

**Anxiety.**    Another strong theme that was described in the data was "anxiety", linked to the Coronavirus Pandemic. This Covid related anxiety manifested itself in many avenues, such as the risk of contracting or spreading Covid-19 in the workplace, as well as bringing it to the individual's own home. The risk and actuality of being redeployed to another region of the service was also described as a significant cause of anxiety by participants. "When the pandemic hit it no longer became possible to leave your work at the door when you came home at night, I never felt safe and I never felt clean. I couldn't relax. I couldn't destress with my family because I was constantly worried about bring covid 19 to work or from work" (ID: 306).

**Poor management and staff driven stress.**    Stress from "poor management" presented itself in several manners, such as pressure being applied by management, conflict with management, and a lack of support from management. This was alluded to by participants in both qualitative responses. A lack of direction, absence of supervision, and management working from home during the pandemic were outlined as significant stressors. "Feeling of being "left to it" by management who were working from home, and knew nothing of the stress on site" (ID: 162). Respondents also outlined feeling undervalued by their manager. "The way staff have been treated in the last year or so has left them feeling unsupported and expendable" (ID: 178)

"Staff driven stress" was strongly detailed in "additional comments" by the participants of this study. It was frequently pointed out that management and peer relationships were the stronger drivers of stress, as opposed to stress induced by client behaviour. Stress linked to peers typically presented itself in the form of poor treatment of clients, lack of team cohesion, and gossiping. "I must stress the service users have are not the frustration it's unrealistic expectations from upper management and HIQA" (ID: 364).

**Service user disruption.**    Another strong theme described in the data was increased stress, linked to the disruption of the Covid-19 restrictions on the routines of service users, this led not only to client confusion, and behaviours that challenge, but many several participants also detailed how this caused client skill deterioration. "Very stressful (for) Service users whom are suffering because of the pandemic, change of routine for them, trying to understand the pandemic, massive changes in their lives making it more difficult for them hence change in the person and behaviour" (ID: 332). It was also indicated that stress on service users from change

of routine caused by the pandemic, at times resulted in physical aggression towards staff. "An increase in physical attacks towards staff, which is stressful, waiting all day to see what triggers it because at this point it's not if, it's when" (ID: 326).

**Leaving the sector.** A common theme described in response to both questions was a desire by many of the participants to leave the "social care worker" sector. Several of the themes previously described were linked to this theme, particularly- staff shortages, poor management, as well as low pay, and feeling underappreciated for the contributions provided as part of their role. "It kills me because I've always wanted to work with people it's always been my calling so many hard years studying and volunteering, 5 years of love and passion for the job and those I support and these last two years have just worn me down to the bone" (ID: 326).

## Discussion

This cross-sectional study, aimed to examine associations between burnout and stressors linked to the role of supporting adults with intellectual disabilities, in a sample of 329 Irish HCWs.

As hypothesised in H1, we observed larger degrees of personal and work-related burnout than client-related burnout. This was a common finding when reviewing the literature [16, 17, 58, 59], with potential contributing factors including increased efforts and workload experienced during the pandemic, as discussed by Adamis et al. [17]. In line with Vassos & Nankervis [28], the organisational stressor "Lack of Resources" had the strongest association with personal burnout (hypothesis H3). Notably, while the stressors "Low-job status" and "Bureaucracy" were both found to have a positive association with work-related burnout, "Lack of resources" was found to have a greater association with work-related burnout than "Low-job status; this does not support hypothesis H4 or the findings by Vassos & Nankervis [28], and it might be due to the fact that participants in our study reported on average low levels of stress related to low-job status. Hypothesis H5, which was informed by Vassos & Nankervis [28], was partially supported, with "Client challenging behaviour" found to have the strongest statistically significant correlation with Client-related burnout. However, despite "Client challenging behaviour" making a statistically significant contribution to the hierarchical multiple regression, both "Lack of resources" and "Poor client skill" were both found to make greater statistically significant contributions. These quantitative findings were supported by the qualitative responses, which outlined an increase in client-related difficulties as well as stress driven by staff, or management generated. Dixon et al. [60] found an association between burnout and an excessive workload being placed on HCWs, particularly when HCWs did not have the opportunity to spend an appropriate amount of time with clients, this hurt the quality of their care.

The correlation and regression analyses showed that "lack of resources" was the stressor most strongly associated with all three types of burnout, particularly personal burnout, independent of socio-demographic or work circumstances. The thematic analysis on the qualitative data aligned with this, finding the theme of "excessive workload" due to the Covid-19 pandemic to be associated with increased stress ("lack of resources" encapsulates many of the same aspects of the theme of "excessive workload"). While "bureaucracy" was also an important stressor for personal and work-related burnout. Although the risks of bureaucracy being a driver of burnout in healthcare settings is outlined by Gunderman & Lynch [61], research studies demonstrating this association in a healthcare setting could not be found when reviewing the literature. "Client challenging behaviour" and "poor client skill" were both found to contribute to higher levels of client-related burnout. This ambiguity into the clients influence in driving burnout was also present in the qualitative data, with the themes of "service user

disruption" and "poor management and staff driven stress" associated with greater stress than behaviours displayed by clients.

Mean levels of personal, work-related, and client-related burnout were found to be notably higher than those of the social care workers from the PUMA Study [16], supporting hypothesis H6. The levels in our study were considerably higher than any other study examined in the literature review [16, 38, 58, 59]. Although little research has been conducted on burnout amongst Irish social care workers, an Irish study by McMahon et al. [38] on staff working with individuals with intellectual disabilities found levels of CBI burnout to be at a higher level than the average of those found in meta-analyses in which the CBI scales were used [18, 19]. This may indicate that Irish frontline staff working in this sector may be in greater jeopardy of developing burnout than in other healthcare professions, and than staff working with adults with intellectual disabilities in other countries. Although identifying the specific reasons for these differences is beyond the scope of this study, it is possible that factors such as underfunding of Irish disability services may play a role in this case [62, 63]. "Lack of resources" was the primary driver of burnout in this study, which may be further evidence of the ties between underfunding and burnout in Irish disability services, given that under resourced services may be likely to place a greater workload on staff.

Staff indicated that the Covid-19 pandemic increased their levels of stress, which supported hypothesis H2. Despite this finding, the influence of the Covid-19 pandemic on stress levels was not found to make a statistically significant contribution to any of the dimensions of burnout in the multiple regression. Despite this, the qualitative data indicated the pandemic intensified organisational stressors that drive burnout, such as anxiety regarding personal health as a result of possibly being exposed to Covid-19, as well as an increase in workload, which aligns with the literature review from Leo et al. [34]. Various studies have outlined increased burnout among HCWs during the pandemic due to increased workload and the stress or anxiety associated with this [30, 31, 59]. Khashne et al. [59] outlined a strong association between levels of anxiety and burnout, although the extent of HCWs workload was not examined in this study. Kamali et al. [30] found a positive link between excessive workload taken on by HCWs and burnout. Similarly, Bellanti et al. [31] indicated that workload was a driver of burnout that led to considerations of participants leaving their job, however, other factors such as HCWs having direct contact with infected individuals was also highlighted as a factor.

Thus, the findings of this study outline the impact that organisational factors had in driving burnout among Irish HCWs supporting adults with intellectual disabilities during the Covid-19 pandemic. The greatest organisational driver of burnout found was "lack of resources"/ "excessive workload", which is likely linked to the additional workload brought about by the pandemic, staff shortages during the pandemic, and to the overall poor funding that has been frequently linked to disability services in Ireland.

## Strengths and limitations

Strengths of this study included the use of standardised scales for burnout and stress with high levels of internal reliability found for the CBI subscales. The study pilot enabled the optimisation of the quality and readability of the survey. Sharing the survey on social media allowed for wider data collection than pen and paper would have provided.

However, a limitation of this study was that although the SSQ subscales were all found to have acceptable internal reliability, they ranged from high internal reliability to poor. These scores were very similar to those found by Hatton et al. [64] and highlight the potential need for better measurements of stress. While the online survey enabled easy access to many prospective participants, the potential selection bias associated with convenience sampling is an

important limitation (discussed above), as demonstrated by the fact that the vast majority of the respondents were women (90.6%), although this may reflect the well-documented predominance of women in social care professions [65]. The participants were also generally young, with about 60% under 40, this sample may therefore not be entirely representative of the population, limiting the generalisability of the study. Thus, the findings need to be considered in light of the characteristics of this sample and the potential selection bias.

## Implications

Given the high levels of burnout observed both in this and a previous study by McMahon et al. [38], it is recommended that future research be conducted among HCWs who work with adults with intellectual disabilities in Ireland and other settings to tease out potential enablers and barriers to wellbeing at work, considering in particular organisational stressors. This is of particular relevance now that the Coronavirus pandemic is over, as an understanding of how burnout and organisational stressors have changed, is required. Given the possibility of another pandemic reoccurring in the future, it is recommended that organisations that support adults with intellectual disabilities consider the findings from this study, and where necessary develop/enhance plans of action that are ready should another pandemic occur [35]. Further qualitative investigations would be important to explore the varied experiences of healthcare staff in different settings and with different organisational circumstances. "Lack of resources" was undoubtedly the stressor associated with the greatest levels of overall burnout in this study. "Lack of resources" is an organisationally generated stressor that possibly indicates underfunding in the intellectual disability sector in Ireland [62, 63]. It is likely for this stressor to be addressed, a suitable level of funding must be invested in respective services and expended appropriately, while appropriate staff contingency plans must also be in place while the pandemic is still impacting services.

An approach that emphasises building an open and constructive relationship between the staff and management may be of significant benefit, in addressing the stressors of "poor staff support", and "poor management and staff driven stress". An organisational intervention that was introduced by the Mayo Clinic (regarded by U.S. News and World Report to have the best hospitals in the world) [66], was the "Listen-Act-Develop" model [67]. This program seeks to prioritize the culture of an organisation by producing a safe environment for staff to identify and reduce stress, as well as build a constructive relationship between the organisation and HCWs [67]. A brief team-based organisational intervention called "Take Care!" was successful in reducing levels burnout by identifying work stressors, and tackling these through "problem solving strategies" [27]. This training also emphasised team communication and social support among colleagues [27]. A meta-analysis on implementing organisational strategies to reduce burnout among HCWs during the Covid-19 pandemic found organisational interventions had a moderate reduction in burnout scores among HCWs [68]. The importance of including the healthcare organisation in strategies and interventions being formulated is stressed by Sultana et al. [25], as a means of communicating and understanding challenges experienced by both staff members and service providing organisations. Future experimental research is warranted to robustly test the potential benefits of interventions that prioritise the reduction of organisational stressors driving burnout, as these may provide a viable way for healthcare organisations to bolster staff, service users and services as a whole [4].

## Conclusions

This survey study investigated associations between multiple stressors and burnout among HCWs supporting individuals with intellectual disabilities. High levels of personal and work-

related burnout were exhibited by the participants of this study, fitting the criteria of "moderate burnout". Client-related burnout was found to be considerably lower than personal burnout, although it was still reasonably high it did not meet the threshold of "moderate burnout". The analyses indicated an important role of organisational stressors for burnout, particularly when working under increased workload conditions during the pandemic. The quantitative and qualitative data were generally aligned in the stressors they outlined as driving burnout. However, despite the Covid-19 pandemic not being found to be a statistically significant driver of burnout, its impact in driving burnout was outlined comprehensively in the qualitative data. Research has demonstrated the efficacy of organisational interventions in reducing burnout among HCWs [25, 68], it is recommended that future experimental research is carried out to develop a further evidence-base for the reduction and prevention of burnout in healthcare settings.

## Supporting information

**S1 Table. Sample characteristics.** Descriptive statistics for socio-demographic characteristics. (DOCX)

**S2 Table. Staff Stressor Questionnaire subscales.** Descriptive statistics for the Stress scales. (DOCX)

**S3 Table. Pairwise Pearson's correlations.** Bivariate correlations between measures. (DOCX)

## Author Contributions

**Conceptualization:** Patrick Clancy, Marica Cassarino.

**Data curation:** Patrick Clancy.

**Formal analysis:** Patrick Clancy, Marica Cassarino.

**Investigation:** Patrick Clancy.

**Methodology:** Patrick Clancy, Marica Cassarino.

**Resources:** Patrick Clancy.

**Software:** Patrick Clancy, Marica Cassarino.

**Supervision:** Marica Cassarino.

**Writing – original draft:** Patrick Clancy.

**Writing – review & editing:** Patrick Clancy, Marica Cassarino.

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
