## [Decision Letter · Decision Letter 0]

8 Jul 2024

PONE-D-24-07726Title: Burnout and Organisational Stressors among Healthcare Staff Working with Adults with Intellectual Disabilities in IrelandPLOS ONE

Dear Dr. Clancy,

Thank you for submitting your manuscript to PLOS ONE. After careful consideration, we feel that it has merit but does not fully meet PLOS ONE’s publication criteria as it currently stands. Therefore, we invite you to submit a revised version of the manuscript that addresses the points raised during the review process.

We look forward to receiving your revised manuscript.

Kind regards,

Soham Bandyopadhyay

Academic Editor

PLOS ONE

Journal Requirements:

Reviewers' comments:

Reviewer's Responses to Questions

**Comments to the Author**

1. Is the manuscript technically sound, and do the data support the conclusions?

Reviewer #1: Yes

Reviewer #2: Yes

2. Has the statistical analysis been performed appropriately and rigorously? 

Reviewer #1: Yes

Reviewer #2: Yes

3. Have the authors made all data underlying the findings in their manuscript fully available?

Reviewer #1: Yes

Reviewer #2: Yes

4. Is the manuscript presented in an intelligible fashion and written in standard English?

Reviewer #1: Yes

Reviewer #2: Yes

5. Review Comments to the Author

Reviewer #1: See review comments

Reviewer comments

Title: Burnout and Organisational Stressors among Healthcare Staff Working with Adults with Intellectual Disabilities in Ireland

This manuscript reports findings from the perspectives of healthcare staff working with adult clients living with intellectual disabilities, with a goal of exploring burnout and the role of organisational stressors. The study is important as it intended to add understanding by extension into previous studies from personal, work and client-related ambits of burnout, as the authors had identified paucity in work related to organisational factors. Thus, the study is significant. However, the paper could benefit from more editing, organisation of content, and more clarifications in various places. Below are detailed comments to help the authors improve on the reporting of this important work.

Abstract:

The abstract is professionally written.

The conclusion is that organisation-focused intervention is needed to address burnout; however, this could flow better by adding more details on the findings.

1. Introduction

This section contains a detailed description and critique of literature, and a strong justification for the study, with identification of knowledge gaps. However, a separate section with a subtitle of study aims and objectives is desirable, to improve the structure.

I am also unsure if the six hypotheses are necessary for this study, considering that the authors have highlighted the challenges of the subject matter and the gaps.

The paper also omitted to address these hypotheses in detail in the discussion section.

2.Materials and Methods

2.1 Design

This study was carried out using an online survey with Irish healthcare staff who work with adults with intellectual disabilities. This section needs more detailed explanation, including literature basis, and a justification for online survey, comparison with other tools, The section could also benefit from literature reference/s.

2.2 Participants

This also needs more detail with literature justification for participant recruitment, comparison with other recruitment tools, ethical considerations, along with appropriate references.

2.3 Materials and Measures

The survey consisted of six parts, including the measures described below.

This subheading and opening sentence are confusing and could contain more detailed description to indicate the contents of the whole section.

Although 2.3.1 Burnout and 2.4.2 Work Stressors contain details of the questionnaires used in the study, the other subheadings in this section including 2.4.3 Covariates, 2.4.4 Qualitative Questions, 2.5 Procedure and 2.6 Data analysis should contain more information and in separate sections.

2.4.3 Covariates

Analyses controlled for the gender of participants, age, level of education, duration of employment in your current role, hours typically worked in a week in this role, work setting, the degree to which the Covid-19 pandemic influenced their stress level (on a 5-point Likert scale of 5), participants reported how often they had worked with a client with a mental health difficulty in the past 6 months (on a 5- point Likert scale).

I would recommend that the authors consider rewriting this section for clarity, including use of italics or quotes, (e.g., duration of employment in your current role)

2.5 Procedure

Data was collected online between the dates of January the 29th 2022 and April the 21st 2022. Before the data collection, the survey was piloted with five individuals who gave feedback on the quality of the questions and any technical issues, these responses were included in the data analyses.

There should also be literature basis for this to support pilot, benefits, etc. Reference/s should also be included.

The heading is also misleading- consider ‘Data collection.’

2.6 Data analysis

Both quantitative and qualitative data was collected as part of this research project. This research project used a deductive-simultaneous design, the core component of this study was quantitative, while the supplemental component was qualitative. The data collected in this study was observational. Data was downloaded from Qualtrics to excel. IBM SPSS 28.0 statistical software was used to carry out the quantitative analysis, while NVivo v.12 was used for qualitative data.

There is no literature basis provided to support this section.

This section is also misleading and could benefit from clarity. I am unsure if the sentence a deductive-simultaneous design is appropriate here under analysis subheading and not under ‘Design.’

I respectfully object to this statement: The data collected in this study was observational, as it is also misleading, considering the tools that were referred to.

Reference materials also need to be added to support the analysis tool indicated.

2.6.1 Quantitative Data Analysis

The first part of this section should contain appropriate reference/s, before Sawada’s.

Also consider adding supportive references for the first few lines of the second paragraph.

2.6.2 Qualitative Data Analysis

The reliance on Braun & Clarke’s guide on thematic analysis should be based on the latest guidance, including reflexivity provisions. Please access the latest revised materials, and include in this section, along with a reflexivity in the discussion section.

3. Results

This section is appropriately detailed and professionally written, with specific figures and comparison with underlying knowledge.

However, this could be presented in greater depth to flow into the Discussion section.

-use of words such as About 60%; Over 90%, etc. in Descriptive statistics should be reconsidered for clarity.

- All the participants worked directly with adults with intellectual disabilities, excluding two participants who work in the management of the service for adults with intellectual disabilities.

- I would recommend that a consideration be given to rewriting this for clarity

4. Discussion

The discussion section could be in greater depth with:

-highlight of the findings with comparison with literature in depth, with appropriate references

-the six hypotheses should be explored more as indicated in the aims of the study (As hypothesised, we observed larger degrees of personal and work-related burnout than client-related burnout. appears to be the only reference to this subject). As indicated earlier, I am unsure if hypothesising would add benefit to this paper and would respectfully recommend that it be reconsidered.

-consider breaking the whole section to include separate subheadings- Strengths and Limitations (which could include some reflection on thematic analysis), Recommendations, Conclusion.

Reviewer #2: Thank you for the invitation to review this paper titled “Burnout and Organisational Stressors among Healthcare Staff Working with Adults with Intellectual Disabilities in Ireland.”

This study investigates the associations between organisational stressors and burnout among healthcare workers supporting adults with intellectual disabilities in Ireland during the COVID-19 pandemic. It uses quantitative and qualitative methods to explore these relationships, providing valuable insights into the factors contributing to burnout in this context.

Major Comments

Scientific Rigor and Methodology

• The study employs a robust cross-sectional survey design with an adequate sample (n=329), enhancing the reliability of the findings.

• Using validated instruments (Copenhagen Burnout Inventory [CBI] and Staff Stressor Questionnaire [SSQ]) is appropriate and methodologically sound.

• The combination of quantitative correlational and qualitative thematic analysis provides a comprehensive understanding of the issues.

Areas for Improvement:

1. While the CBI and SSQ are established tools, additional information on their validation in the context of the Irish healthcare system and during the COVID-19 pandemic would strengthen the methodology. Please consider including more details on the psychometric properties of these instruments in this specific study context.

2. The data availability statement currently includes conflicting information. It should be clarified to indicate that all data are fully available without restriction and provide a link to a publicly accessible repository where the data can be accessed (i.e., https://osf.io/ctpmb/). This will ensure compliance with PLOS ONE’s guidelines.

Data and Results Presentation

• The results are well presented using tables to illustrate key findings.

• The hierarchical multiple regression analysis effectively identifies the strongest predictors of burnout, and the qualitative results provide rich, complementary insights.

Areas for Improvement:

3. Consider including some of the supplementary tables as main tables within the manuscript. This will enhance the readability and accessibility of critical findings. For example, tables detailing demographic characteristics, burnout scores, and predictor variables could be moved to the main text.

4. While the thematic analysis is well-executed, providing more direct quotes from participants could enhance the richness and authenticity of the qualitative findings.

Interpretation and Implications

• The discussion effectively interprets the findings in the context of existing literature, highlighting the significance of "lack of resources" as a key stressor.

• The implications for organisational interventions to reduce burnout are thoughtfully considered.

Areas for Improvement:

5. Expand the discussion on the broader implications of the findings, particularly regarding the ongoing and future impact of pandemics on healthcare workers. This could provide additional relevance and context for the study’s findings.

6.While recommendations for future research are provided, consider specifying potential methodologies or areas of focus that could build on the current study’s findings.

Minor Comments

7. The introduction provides a comprehensive background and clearly states the research gap and objectives. To engage a broader audience, consider briefly mentioning the significance of this research in the context of global health crises early in the introduction.

8. The methods section is detailed and transparent. To maintain clarity for all readers, ensure that any abbreviations used are defined upon first use.

9. The discussion is well-structured, but ensure that all key findings are succinctly summarised in the conclusion section to reinforce the study's main messages.

6. PLOS authors have the option to publish the peer review history of their article (what does this mean?). If published, this will include your full peer review and any attached files.

Reviewer #1: No

Reviewer #2: No

---

## [Author Response · Author response to Decision Letter 0]

21 Aug 2024

PONE-D-24-07726

Reviewer Comments and Responses

Dear Editor, 

We would like to extend our sincerest thanks to you and the Reviewers for reviewing our article and giving us the opportunity to revise our manuscript. We are now submitting version R1 with tracked changes and a point-by-point response to all comments, which can be found at the bottom of this letter. Please note that page numbers refer to the manuscript with tracked changes. 

All authors have read and approved the final revised version of the manuscript.

Please accept our sincere thanks for considering our submission. 

Yours, sincerely,

Patrick Clancy

Editor comments

E1. Please ensure that your manuscript meets PLOS ONE's style requirements, including those for file naming. The PLOS ONE style templates can be found at 

Response: We thank the Editor for providing these guides. We have now amended the manuscript to align with the formatting requirements. 

Response: We have amended the data availability statement in the manuscript to indicate the OSF location where the data is publicly accessible (link: https://osf.io/ctpmb) 

3. Please include captions for your Supporting Information files at the end of your manuscript, and update any in-text citations to match accordingly. Please see our Supporting Information guidelines for more information: http://journals.plos.org/plosone/s/supporting-information .

Response: Thank you for this. We have now created a full list of captions for the supporting information files. This can be found at p.40 of the tracked revised manuscript. 

Reviewer 1 comments

R1C1: The conclusion is that organisation-focused intervention is needed to address burnout; however, this could flow better by adding more details on the findings.

Response: Thank you for this point. We have amended this part of the abstract as follows (see bottom of p.2): 

“Our findings highlight an important association between organisational stressors and burnout among frontline staff, suggesting the potential benefit of designing organisationally focused interventions to reduce stress and promote staff wellbeing.”

R1C2: This section contains a detailed description and critique of literature, and a strong justification for the study, with identification of knowledge gaps. a) However, a separate section with a subtitle of study aims and objectives is desirable, to improve the structure.

I am also unsure if the six hypotheses are necessary for this study, considering that the authors have highlighted the challenges of the subject matter and the gaps.

b) The paper also omitted to address these hypotheses in detail in the discussion section.

Response: We thank the Reviewer for this comment. We have created a sub-section in the introduction titled “Study Aims” (see p.7) to enhance legibility. While we appreciate that the study tested a number of hypotheses, we feel that these are appropriately justified in light of previous literature and enabled a thorough examination of the topic. To address point “b” of the comment, we have made it more evident in the discussion section (see p.25-26) where hypotheses were considered and whether these were supported or rejected. 

R1C3: 2.1 Design - This study was carried out using an online survey with Irish healthcare staff who work with adults with intellectual disabilities. This section needs more detailed explanation, including literature basis, and a justification for online survey, comparison with other tools, The section could also benefit from literature reference/s.

Response: We have expanded the Design section to better justify the choice of a survey and data collection, with relevant references (p.9). The limitations of the design are discussed in the limitations section in the discussion (pp.28-29). 

R1C4: 2.2 Participants - This also needs more detail with literature justification for participant recruitment, comparison with other recruitment tools, ethical considerations, along with appropriate references.

Response: We have now provided a description of the supporting information files at p.41. All in -text citations have been updated to the journal style. Participants and ethical information have been updated at p.10 in line with Reviewers’ comments. 

R1C5: 2.3 Materials and Measures - The survey consisted of six parts, including the measures described below. This subheading and opening sentence are confusing and could contain more detailed description to indicate the contents of the whole section.

Although 2.3.1 Burnout and 2.4.2 Work Stressors contain details of the questionnaires used in the study, the other subheadings in this section including 2.4.3 Covariates, 2.4.4 Qualitative Questions, 2.5 Procedure and 2.6 Data analysis should contain more information and in separate sections.

Response: We have improved the “Materials and measures” section to give further details about the structure of the survey and the types of questions asked. These changes can be found at pp.11-14. 

R1C6: 2.4.3 Covariates - Analyses controlled for the gender of participants, age, level of education, duration of employment in your current role, hours typically worked in a week in this role, work setting, the degree to which the Covid-19 pandemic influenced their stress level (on a 5-point Likert scale of 5), participants reported how often they had worked with a client with a mental health difficulty in the past 6 months (on a 5- point Likert scale). I would recommend that the authors consider rewriting this section for clarity, including use of italics or quotes, (e.g., duration of employment in your current role).

Response: As suggested, we amended this section (pp.13) to enhance clarity about the measured used. 

R1C7 - 2.5 Procedure - Data was collected online between the dates of January the 29th 2022 and April the 21st 2022. Before the data collection, the survey was piloted with five individuals who gave feedback on the quality of the questions and any technical issues, these responses were included in the data analyses.

There should also be literature basis for this to support pilot, benefits, etc. Reference/s should also be included. The heading is also misleading- consider ‘Data collection.’

Response: Thank you for these comments. Piloting a survey is a standard good practice procedure that is highly recommended to ensure the appropriateness of the content and order of the questions in the survey. We have now expanded the justification of this approach at p.15. Similarly, the heading “Procedure” is a standard type of heading label for quantitative articles, but we appreciate that the meaning may vary across contexts and we have thus specified “Data Collection procedure” 

R1C8 - 2.6 Data analysis Both quantitative and qualitative data was collected as part of this research project. This research project used a deductive-simultaneous design, the core component of this study was quantitative, while the supplemental component was qualitative. The data collected in this study was observational. Data was downloaded from Qualtrics to excel. IBM SPSS 28.0 statistical software was used to carry out the quantitative analysis, while NVivo v.12 was used for qualitative data. There is no literature basis provided to support this section.

Response: We have now added references to this section where deemed appropriate (see pp.14-15). 

R1C9: This section is also misleading and could benefit from clarity. I am unsure if the sentence a deductive-simultaneous design is appropriate here under analysis subheading and not under ‘Design.’

Response: Thank you for this. We have moved the mention of the deductive-simultaneous design to the design section, with relevant reference (see p.9). We have updated the data analysis section to enhance clarity. 

R1C10 - I respectfully object to this statement: The data collected in this study was observational, as it is also misleading, considering the tools that were referred to.

Response: We used the term “Observational” drawing from Epidemiology, where an observational quantitative study is a study using numbers to measure variables without IV manipulation (see for instance https://www.ncbi.nlm.nih.gov/pmc/articles/PMC6970097/). However, we appreciate that this term may cause confusion across different disciplines, and we have thus removed it.

R1C11 - Reference materials also need to be added to support the analysis tool indicated.

Response: We have included references to the software used for the analysis at p.14

R1C12: 2.6.1 Quantitative Data Analysis - The first part of this section should contain appropriate reference/s, before Sawada’s. Also consider adding supportive references for the first few lines of the second paragraph.

Response: We thank the reviewer for this point and added references where deemed appropriate. 

R1C13 - 2.6.2 Qualitative Data Analysis - The reliance on Braun & Clarke’s guide on thematic analysis should be based on the latest guidance, including reflexivity provisions. Please access the latest revised materials, and include in this section, along with a reflexivity in the discussion section.

Response: We have updated the Braun and Clarke’s reference at p.16, and added at p.17 a section on reflexivity as applied to this study

R1C14: 3. Results - This section is appropriately detailed and professionally written, with specific figures and comparison with underlying knowledge. However, this could be presented in greater depth to flow into the Discussion section.

-use of words such as About 60%; Over 90%, etc. in Descriptive statistics should be reconsidered for clarity.

- All the participants worked directly with adults with intellectual disabilities, excluding two participants who work in the management of the service for adults with intellectual disabilities.

- I would recommend that a consideration be given to rewriting this for clarity.

Response: We thank the Reviewer for this suggestion. We have amended the sample characteristics section at pp. 17-18 to make the percentages accurate and improve the flow of the narrative synthesis. Following a suggestion from Reviewer 2, we have also incorporated some of the supplementary tables into the results section to enhance accessibility to the data. 

The discussion section could be in greater depth with:

R1C15: -highlight of the findings with comparison with literature in depth, with appropriate references

Response: Thank you for this comment. We have completed a major update of the Discussion section to better link the findings to our hypotheses and relevant literature. These changes can be seen at pp.25-28. 

R1C16: -the six hypotheses should be explored more as indicated in the aims of the study (As hypothesised, we observed larger degrees of personal and work-related burnout than client-related burnout. appears to be the only reference to this subject). As indicated earlier, I am unsure if hypothesising would add benefit to this paper and would respectfully recommend that it be reconsidered.

Response: Thank you for suggestion, as it adds greater clarity to the discussion. All six hypothesis are now explicitly flagged in the discussion section (see pp.25-28). 

R1C17: -consider breaking the whole section to include separate subheadings- Strengths and Limitations (which could include some reflection on thematic analysis), Recommendations, Conclusion.

Response: We thank the reviewer for this valuable suggestion. We have now added the sub-headings “strengths and limitations”, “implications” and “conclusion”. 

Reviewer #2 comments

Thank you for the invitation to review this paper titled “Burnout and Organisational Stressors among Healthcare Staff Working with Adults with Intellectual Disabilities in Ireland.”

This study investigates the associations between organisational stressors and burnout among healthcare workers supporting adults with intellectual disabilities in Ireland during the COVID-19 pandemic. It uses quantitative and qualitative methods to explore these relationships, providing valuable insights into the factors contributing to burnout in this context.

Major Comments

Scientific Rigor and Methodology

• The study employs a robust cross-sectional survey design with an adequate sample (n=329), enhancing the reliability of the findings.

• Using validated instruments (Copenhagen Burnout Inventory [CBI] and Staff Stressor Questionnaire [SSQ]) is appropriate and methodologically sound.

• The combination of quantitative correlational and qualitative thematic analysis provides a comprehensive understanding of the issues.

Response: We thank the Reviewer for these positive and supportive comments. 

Areas for Improvement:

R2C1. While the CBI and SSQ are established tools, additional information on their validation in the context of the Irish healthcare system and during the COVID-19 pandemic would strengthen the methodology. Please consider including more details on the psychometric properties of these instruments in this specific study context.

Response: Thank you for this. We have included additional information on psychometric properties for the CBI and for the SSQ at pp. 11 and 12 

R2C2. The data availability statement currently includes conflicting information. It should be clarified to indicate that all data are fully available without restriction and provide a link to a publicly accessible repository where the data can be accessed (i.e., https://osf.io/ctpmb/). This will ensure compliance with PLOS ONE’s guidelines.

Response: Thank you for flagging this inconsistency. We have amended the data availability statement in the manuscript to indicate the OSF location where the data is publicly accessible. 

R2C3. Consider including some of the supplementary tables as main tables within the manuscript. This will enhance the readability and accessibility of critical findings. For example, tables detailing demographic characteristics, burnout scores, and predictor variables could be moved to the main text.

Response: We have included 4 of the supplementary tables to aid the narrative synthesis. These are numbered as Table 1 to Table 4 and can be found in the results section. 

R2C4. While the thematic analysis is well-executed, providing more direct quotes from participants could enhance the richness and authenticity of the qualitative findings.

Response: Thank you for this comment. We have included further quotes from the qualitative data at pp.22-25.

• The discussion effectively interprets the findings in the context of existing literature, highlighting the significance of "lack of resources" as a key stressor.

• The implications for organisational interventions to reduce burnout are thoughtfully considered.

R2C5. Expand the discussion on the broader implications of the findings, particularly regarding the ongoing and future impact of pandemics on healthcare workers. This could provide additional relevance and context for the study’s findings.

Response: Thank you for this point. We have created an Implications section (p. 29) where we have enhanced the discussion on the potential impacts of the present findings. 

R2C6.While recommendations for future research are provided, consider specifying potential methodologies or areas of focus that could build on the current study’s findings.

Response: Thank you. We have expanded the implications section (pp. 29) to add considerations about future research focus and methods. 

R2C7. The introduction provides a comprehensive background and clearly states the research gap and objectives. To engage a broader audience, consider briefly mentioning the significance of this research in the context of global health c

---

## [Decision Letter · Decision Letter 1]

31 Oct 2024

Title: Burnout and Organisational Stressors among Healthcare Staff Working with Adults with Intellectual Disabilities in Ireland

PONE-D-24-07726R1

Dear Dr. Clancy,

We’re pleased to inform you that your manuscript has been judged scientifically suitable for publication and will be formally accepted for publication once it meets all outstanding technical requirements.

Kind regards,

Ramona Bongelli, Ph.D.

Academic Editor

PLOS ONE

Additional Editor Comments (optional):

Reviewers' comments:

Reviewer's Responses to Questions

**Comments to the Author**

1. If the authors have adequately addressed your comments raised in a previous round of review and you feel that this manuscript is now acceptable for publication, you may indicate that here to bypass the “Comments to the Author” section, enter your conflict of interest statement in the “Confidential to Editor” section, and submit your "Accept" recommendation.

Reviewer #2: All comments have been addressed

2. Is the manuscript technically sound, and do the data support the conclusions?

Reviewer #2: Yes

3. Has the statistical analysis been performed appropriately and rigorously? 

Reviewer #2: Yes

4. Have the authors made all data underlying the findings in their manuscript fully available?

Reviewer #2: Yes

5. Is the manuscript presented in an intelligible fashion and written in standard English?

Reviewer #2: Yes

6. Review Comments to the Author

Reviewer #2: Thank you for the opportunity to review the revised manuscript. I am pleased to note that all comments have been adequately addressed, improving the paper's quality and clarity. I appreciate your diligence and wish you success in your future research endeavours.

7. PLOS authors have the option to publish the peer review history of their article (what does this mean?). If published, this will include your full peer review and any attached files.

Reviewer #2: No

---

## [Editor Report · Acceptance letter]

11 Nov 2024

PONE-D-24-07726R1 

PLOS ONE

Dear Dr. Clancy, 

I'm pleased to inform you that your manuscript has been deemed suitable for publication in PLOS ONE. Congratulations! Your manuscript is now being handed over to our production team.

Kind regards, 

on behalf of

Professor Ramona Bongelli 

Academic Editor

PLOS ONE